# Perspectives on dental health and oral hygiene practice from US adolescents and young adults during the COVID-19 pandemic

Long Zhang[1], Marika Waselewski[2], Jack Nawrocki[3], Ian Williams[4], Margherita Fontana[1], Tammy Chang[2,5]*

1 School of Dentistry, University of Michigan, Ann Arbor, MI, United States of America, 2 Department of Family Medicine, University of Michigan, Ann Arbor, MI, United States of America, 3 College of Literature, Science & the Arts, University of Michigan, Ann Arbor, MI, United States of America, 4 School of Dental Medicine, University of Connecticut, Farmington, CT, United States of America, 5 Institute for Healthcare Policy & Innovation, University of Michigan, Ann Arbor, MI, United States of America

* tachang@med.umich.edu

## Abstract

### Background

Adolescence is a critical time for adopting health behaviors which continue through adulthood. There is a lack of data regarding perspectives of US adolescents and young adults on their dental health and oral hygiene practice.

### Methods

Adolescents and young adults, age 14–24, from MyVoice, a nationwide text message poll of youth. were asked five open-ended questions on the importance of dental health and impact of the COVID-19 pandemic. Responses were qualitatively analyzed using thematic analysis. Chi-square test was used to examine differences in experiences by demographics.

### Results

Of 1,148 participants, 932 responded to at least one question. The mean age was 19 years. Respondents were largely male (49.5%) and non-Hispanic white (62.4%). Most (92%) respondents perceived dental health as important or somewhat important and emphasized overall dental health and hygiene (38.6%) and aesthetics (18.3%). About half (49.2%) of respondents stated they have had at least one cavity since middle school. Just over half (54.8%) reported brushing and flossing to care for their dentition. 58% visited a dentist at least every 6 months, while 38% visited a dentist less frequently or not at all. Being non-cisgender, non-Hispanic black, Hispanic, and receipt of free or reduced lunch was associated with less frequent dental visits. 44% stated COVID-19 impacted their dental health, with many mentioning scheduling difficulties or worsened dental hygiene.

**Data Availability Statement:** Our work uses data from MyVoice (www.hearmyvoicenow.org) a national text message survey of youth (ages 14–

24). As part of our IRB approval and protections for this vulnerable population, we require Data Sharing Agreements to be executed with any individual or organization interested in accessing our data. Researchers interested in this data can contact the authors or the University of Michigan Medical School IRB (irbmed@umich.edu) for more information.

**Funding:** TC received funding for this research from the Michigan Institute for Clinical & Health Research and the University of Michigan Department of Family Medicine. The funders had no role in study design, data collection and analysis, decision to publish, or preparation of the manuscript.

**Competing interests:** The authors have declared that no competing interests exist.

## Conclusions

Most youth in our study consider dental health important, though their oral hygiene practice may not follow ADA guidelines and self-reported dental caries are high. Dental healthcare among youth has been affected by the COVID-19 pandemic with interruption in regular dental visits and changes in hygiene habits. Re-engagement of adolescents and young adults by dental care providers via greater access to appointments and youth-centered messaging reinforcing hygiene recommendations may help youth improve dental health now and in the future.

## Introduction

Oral health is an essential part of the overall health and well-being of an individual. Dental caries (or tooth decay, which can result in painful cavities in teeth) in children remain the primary oral health challenge in the United States [1] and the world [2]. Untreated tooth decay can result in pain and infection, which may affect an individual's eating, speaking, playing, and learning. Poor oral health in children is linked to school absence and worse academic performance [3].

Adolescents represent a unique and often overlooked group of children who experience many biological, developmental, and social transitions. Adolescence is a critical time of life for adopting health behaviors which continue through adulthood. According to the oral health surveillance report in 2011–2016 from CDC, more than half (57%) of adolescents aged 12–19 years experienced dental caries. One in six adolescents aged 12–19 years had untreated tooth decay [4]. There has been no detectable improvement in disease burden of dental caries in adolescents in the past two decades in the United States [5]. However, even though dental caries remains highly prevalent in adolescence, this period of life has been largely neglected in oral health research [5].

Previous studies attributed the high level of dental caries during adolescence to an increase in susceptible newly erupted permanent tooth surfaces, high carbohydrate dietary habits, independence to seek or avoid dental care, a low priority for oral hygiene, and additional social factors [6]. The American Academy of Pediatric Dentistry (AAPD) recommends fluoride, oral hygiene, diet management, and sealant placement as the primary prevention strategies to control dental caries in adolescents [6]. Yet the success of these preventive efforts depends on how adolescents perceive the importance of their dental health, maintain oral hygiene routine, and follow up with dental health care professionals. Also, in the context of COVID-19 pandemic, there is a lack of data regarding adolescents' perspectives of impact of the COVID-19 pandemic on their dental health.

The aim of our study was to understand the perspectives of a nationwide sample of adolescents and young adults regarding the importance of their dental health, experiences with dental caries, how they care for their dental health, how often they visit a dentist, and the impact of COVID-19 pandemic on their dental health.

## Methods

Data from five open-ended questions were obtained from the MyVoice cohort. MyVoice is an ongoing longitudinal nationwide text message poll that seeks to understand youth opinions on salient issues related to health and policy. Participants are a diverse sample of over 1,000 youth aged 14–24 years; details on study procedures have been previously described elsewhere [7].

Briefly, participants' age, gender, race/ethnicity, education level, census region of residence, and free lunch status were collected upon consent for the study. This study was approved by the University of Michigan Institutional Review Board with online written consent completed by all participants and a waiver of parental consent for minors as the study was deemed minimal risk.

Questions were sent to 1,148 MyVoice participants on May 21, 2021 and were iteratively developed by a research team of youth, physicians, dentists, and mixed-methods research experts. Participants were sent the following five questions, which aimed to assess participants' beliefs surrounding the importance of dental health and the impact of the COVID-19 pandemic on their dental care experience:

- How important is dental health to you? Why?

- Since middle school, have you had any cavities? Tell us about it!

- What do you do to take care of your teeth?

- How often do you go to the dentist? What for?

- How has the COVID-19 pandemic affected your dental health?

The survey responses were qualitatively analyzed using thematic analysis. A codebook was iteratively developed by the research team by first reviewing several hundred responses and identifying major concepts or codes. Codes were then defined with illustrative examples to complete the codebook for each question. Two independent researchers reviewed each question and applied codes to each response. Discrepancies between coders were identified and discussed to reach a consensus. Summary statistics were calculated using Microsoft Excel (2016). Chi-square test was used to analyze the difference of selected outcomes by demographics with SAS 9.4 software (Cary, NC, USA).

## Results

Of 1,148 participants, 932 responded to at least one question (response rate = 81.2%). The mean age of respondents was 19 years old (standard deviation: 2 years). Respondents were largely male (49.5%), non-Hispanic white (62.4%), had an education level of some college or technical school (41.0%), resided in the Midwest (32.3%), and did not receive free or reduced lunch (61.2%), a measure of low socioeconomic status (Table 1).

Overall, 92.0% (852/926) considered dental health important or somewhat important to them. The most cited reasons why dental health is important included overall dental health and hygiene (38.6%) and aesthetics (18.3%). Participants who reported on the importance of overall dental health and hygiene noted "it's important to have good dental hygiene", they have "only one set permanent teeth", and "we need our teeth to eat, bite, and talk." (Table 2). The proportion of people who perceived dental health important or somewhat important did not significantly vary by gender, race, or free or reduced lunch receipt (Table 3).

In our sample, 49.2% (442/899) respondents reported having had at least one cavity since middle school. Among all respondents, 9.9% (89/899) specified having one cavity, 28.3% (254/899) reported multiple cavities, and 11.2% (101/899) reported an unspecified number of cavities. The percent of respondents who reported having had at least one cavity significantly varied by gender but did not vary by race or free or reduced lunch receipt. Among those who had tooth decay, 24.4% (108/442) said they received dental treatment for tooth decay. Some respondents attributed their cavity to poor oral hygiene (9.5%, 42/442), poor diet (6.3%, 28/442), a

**Table 1. Respondent demographic characteristics (n = 932).**

| Demographic characteristic | n (%) or mean (std) |
|---|---|
| Age | 19.3 (2.4) |
| 14–17 | 232 (24.9) |
| 18–24 | 700 (75.1) |
| Gender | |
| Male | 461 (49.5) |
| Female | 375 (40.2) |
| Other | 96 (10.3) |
| Race and Ethnicity | |
| Non-Hispanic White | 580 (62.4) |
| Non-Hispanic Black | 63 (6.8) |
| Hispanic | 95 (10.2) |
| Non-Hispanic Other | 192 (20.7) |
| Education level | |
| Less than high school* | 243 (26.1) |
| High school graduates | 136 (14.6) |
| Some college or technical school | 382 (41.0) |
| Associate's degree or technical graduate | 29 (3.1) |
| Bachelor's degree or higher | 142 (15.2) |
| Region | |
| Midwest | 298 (32.3) |
| Northeast | 168 (18.2) |
| South | 253 (27.4) |
| West | 203 (22.0) |
| Received free or reduced lunch | |
| Yes | 359 (38.8) |
| No | 567 (61.2) |

*includes respondents still in high school

health condition or a genetic predisposition (6.1%, 27/442), not visiting a dentist (2.5%, 11/442) or not being able to afford visiting a dentist (1.8%, 8/442). Some respondents revealed that having a cavity caused pain (3.4%, 15/442) or led to an advanced dental treatment such as a crown, root canal treatment, or extraction (3.2%, 14/442).

Among the 883 who answered the question on how they take care of their teeth, 97.2% (858/883) reported brushing and 56.1% (495/883) reported flossing. Over half (54.8%; 484/883) noted that they both brush and floss to care for their teeth, though only 19.1% (169/883) described habits that specifically met the ADA guidelines of brushing twice daily and flossing daily. Other ways respondents reported caring for their teeth included using mouthwash (24.8%), visiting a dentist (10.8%), and avoiding damaging foods like high sugar items such as candies (5.5%).

Most youth (58.4%, 513/878) reported visiting a dentist every 6 months or more frequently, while some youth (37.6%, 330/878) went to a dentist less frequently than every 6 months. Of note, 10.4% (91/878) never visited a dentist for dental care. Common reasons for visiting a dentist included checkups (75.4%, 662/878), dental emergency (3.3%, 29/878), restoration (2.6%, 23/878), and orthodontics (2.3%, 20/878). There was a significant difference in the rate of respondents visiting a dentist every 6 months or more frequently based on gender, race, and status of receiving free or reduced lunch.

**Table 2. Summary of results of thematic analysis.**

| Question, code | n (%) | Example quote |
|---|---|---|
| **How important is dental health to you? Why? (n = 926)** | | |
| *Level of Importance* | | |
| Important | 776 (83.8) | "Very important" "Important enough" |
| Somewhat | 76 (8.2) | "Semi important" "Moderately important" |
| Not Important | 50 (5.4) | "Not that important" "Not at all" |
| *Reasons Important* | | |
| Overall Dental Health and Hygiene | 357 (38.6) | "...it's important to have good dental hygiene" "...only one set permanent teeth" "...we need our teeth to eat, bite, and talk..." "...I want to have healthy teeth later in life" |
| Aesthetics | 169 (18.3) | "...a smile communicates a lot" "Smiles are how I make a good first impression" |
| Prevention | 107 (11.6) | "...without dental health other health problems can occur" "...I care about how I look and reducing risks" |
| Personal experience w/poor dental health | 65 (7.0) | "I've had dental issues so I take it serious" "I have very sensitive gums/teeth" |
| To avoid the dentist | 46 (5.0) | "I also hate going to the dentist for cavity fillings so it motivates me to brush and floss well" "...because dental issues are expensive and painful" |
| *Reasons Unimportant* | | |
| Not top priority | 68 (7.3) | "...I value it but it's not a top priority" "Not too high on my priority list, but still important" |
| No issues | 15 (1.6) | "I don't have issues, so I don't seek care" "I've never had issues with my teeth" |
| **Since middle school, have you had any cavities? Tell us about it! (n = 899)** | | |
| Yes | 442 (49.2) | |
| One | 89 (9.9) | "Yes I've had one and it was a painless experience" "I've had one cavity my whole life" |
| Multiple | 252 (28.0) | "Yes, I actually just had 3 filled, it was fine" "Yes, I had a few small ones that were filled!" |
| Not specified | 101 (11.2) | "Yes" |
| No | 457 (50.8) | "No" "I've never had a cavity and I'm pretty proud of that." |
| *Cavity Experience* (n = 442) | | |
| Received treatment | 108 (24.4) | "i had one cavity, i got it filled by my regular dentist" "yes. i've had a couple of cavities filled throughout the years." |
| Caused by poor hygiene | 42 (9.5) | "Yes, a few because I didn't floss in middle school or high school" "Lots—my parents didn't teach us dental hygiene so my husband has been a huge help" |
| Cause by poor diet | 28 (6.3) | "YES! A lot. I ate candy and soda as a kid and I guess it adds up" "Yes because of junk food" |
| Related to health condition | 27 (6.1) | "A tiny one on the top back molar bc I had acid reflux and needed to start medicine for it" "I have deep ridges in my teeth and tend to mouth breathe which makes me more susceptible to cavities" |
| Didn't visit dentist | 11 (2.5) | "Yes I've had many cavities because I was too poor to afford dental care" "Yes but I didn't need to go to the dentist for them" |
| Cavity caused pain | 15 (3.4) | "Yes I had a few it wasnt the best thing ever it was the worst because it gave you a toothache and headache" "Yes, I've had a couple. It caused me a lot of pain, and those experiences really motivated me to take care of my teeth to a high extent." |

*(Continued)*

**Table 2.** (Continued)

| Question, code | n (%) | Example quote |
|---|---|---|
| Cavity progressed | 14 (3.2) | "One of my teeth rotted and I had to get a crown" "I've had many cavities, a root canal, and recently dental implants" |
| **What do you do to take care of your teeth? (n = 883)** | | |
| Brushing | 858 (97.2) | "i brush my teeth twice a day and go to the dentist regularly" |
| Flossing | 495 (56.1) | "Brush 2x/day and use mouthwash. Occasionally floss" |
| Mouthwash | 219 (24.8) | "I brush twice daily and use mouth wash and visit dentist twice a year" |
| Visit the dentist | 95 (10.8) | "Floss and brush and clean my tongue and go to dentist" |
| Avoiding damaging foods | 49 (5.5) | "Brush my teeth, floss, don't eat or drink a lot of sugar" |
| Nothing | 8 (0.9) | "I don't do anything to take care of my teeth" "No that much" |
| **How often do you go to the Dentist? What for? (n = 878)** | | |
| *Visit Frequency* | | |
| Every 6 months or more frequently | 513 (58.4) | "Twice a year for my checkup and cleaning" |
| Less frequently than every 6 months | 330 (37.6) | "Once every two years usually for a check up" |
| *Reasons for Visits* | | |
| Checkup | 662 (75.4) | "Every 6 months for a checkup" |
| Emergency | 29 (3.3) | "I try to go to the dentist once a year or whenever I have pain in my teeth" |
| Restoration | 23 (2.6) | "At least 2 times a year and for cleanings. And if I need to go back for a cavities" |
| Orthodontics | 20 (2.3) | "Once a month for my braces" |
| **How has the COVID-19 pandemic affected your dental health? (n = 864)** | | |
| No impact | 488 (56.5) | "It has not changed" |
| Impact | 376 (43.5) | "I did not go to the dentist for a year and I put off getting my wisdom teeth removed" "I missed an appointment, but I've since gotten back on track" |
| *Type of Impact* | | |
| Scheduling | 267 (30.9) | "I have to wait an extra couple mouths for my appointment. . ." "I have had to delay dental appointments" |
| Worsened hygiene | 62(7.2) | "I have brushed my teeth less" "I have brush my teeth a lot less and have gotten more cavities due to eating more" |
| Improved dental health | 24 (2.8) | "If anything, it has gotten better" "I think it's made it better and more self aware" |
| Safety concerns | 16 (1.9) | "I didn't go as frequently due to fear of it being unsafe" |
| Eating habits | 9 (1.0) | "Improved it, eating out less and more time to care for my hygiene" |

Regarding whether the COVID-19 pandemic affected respondent's dental health, 56.5% (488/864) said the pandemic had no impact, and 43.5% (376/864) indicated an impact. The types of impact reported by youth included difficulty scheduling appointments (30.9%), worsened hygiene (7.2%), improved dental health (2.8%), safety (1.9%), and eating habits (1.0%).

## Discussion

A majority (92%) of adolescents and young adults surveyed, perceived dental health as important or somewhat important to them regardless of demographics, and many of them

**Table 3. Comparisons of outcomes between demographics.**

| | How important is dental health to you? | | |
|---|---|---|---|
| | Important or somewhat important, n (%) | not important, n (%) | P-value |
| Female | 344 (92.2) | 29 (7.8) | 0.188 |
| Male | 426 (92.8) | 33 (7.2) | |
| Other | 82 (87.2) | 12 (12.8) | |
| Non-Hispanic White | 521 (90.6) | 54 (9.4) | 0.140 |
| Non-Hispanic Black | 57 (90.5) | 6 (9.5) | |
| Hispanic | 89 (94.7) | 5 (5.3) | |
| Non-Hispanic Other | 183 (95.3) | 9 (4.7) | |
| Received free or reduced lunch | 321 (90.4) | 34 (9.6) | 0.175 |
| Not received free or reduced lunch | 525 (92.9) | 40 (92.9) | |
| | Since middle school, have you had any cavities? | | |
| | Had cavities, n (%) | No cavities, n (%) | |
| Female | 202 (55.6) | 161 (44.4) | <0.001 |
| Male | 181 (40.6) | 265 (59.4) | |
| Other | 59 (65.6) | 31 (34.4) | |
| Non-Hispanic White | 269 (48.4) | 287 (51.6) | 0.667 |
| Non-Hispanic Black | 35 (56.4) | 27 (43.6) | |
| Hispanic | 46 (50.6) | 45 (49.4) | |
| Non-Hispanic Other | 91 (48.4) | 97 (51.6) | |
| Received free or reduced lunch | 174 (51.0) | 167 (49.0) | 0.440 |
| Not received free or reduced lunch | 267 (48.4) | 285 (51.6) | |
| | How often do you go to the Dentist? | | |
| | Visit a dentist every 6 months or more frequently, n (%) | Visit a dentist less frequently than every 6 months, n (%) | |
| Female | 209 (60.1) | 139 (39.9) | 0.002 |
| Male | 267 (60.7) | 173 (39.3) | |
| Other | 37 (41.1) | 53 (58.9) | |
| Non-Hispanic White | 342 (62.9) | 202 (37.1) | 0.001 |
| Non-Hispanic Black | 27 (44.3) | 34 (55.7) | |
| Hispanic | 41 (46.1) | 48 (53.9) | |
| Non-Hispanic Other | 102 (56.0) | 80 (44.0) | |
| Received free or reduced lunch | 175 (51.6) | 164 (48.4) | 0.001 |
| Not received free or reduced lunch | 336 (63.0) | 197 (37.0) | |

emphasized the importance of overall dental health and hygiene and aesthetics. This is consistent with a previous report that found adolescents considered dental appearance and oral self-care as important in terms of social interaction with peers, personal and career satisfaction, and future success in social and work life [8]. These positive beliefs in oral health among adolescents may be associated with better oral health outcomes. Broadbent et al found that individuals who held stable favorable dental beliefs from adolescence through adulthood had fewer tooth loss due to caries, less periodontal disease, better oral hygiene, and better self-rated oral health [9].

Despite having positive perspectives in oral health, our study participants (aged 14–24 years) revealed a high prevalence (49%) of self-reported dental caries since middle school, which is consistent with the CDC data that reported 57% prevalence of caries among adolescents aged 12–19 years [4]. This high prevalence of dental caries may be due to a host of factors that were not measured in our study including, the rate of caries experienced during early childhood, increased exposure to cariogenic bacteria, high frequency of sugar consumption, inadequate salivary composition or flow, delayed or insufficient fluoride exposure, and poor oral hygiene [10, 11]. Adolescent's positive beliefs in oral health are also subject to change and the stability of dental health beliefs, rather than a snapshot, may be more predictive of self-reported oral health outcomes [9].

Females and those who identify as "other" gender also reported a statistically significant higher rate of having cavities than male respondents. But there was no significant difference of the rate of having cavities by race and receipt of free or reduced lunch (a proxy for socioeconomic status). It is known that socioeconomic factors such as race and household income level contribute to the disparity of dental caries in U.S. adolescents [4]. The reason why our study did not show the same pattern could result from a different definition of outcome, sampling method, and study period. The lack of associations between self-reported dental caries and socioeconomic status in our study warrants further investigation.

The way in which adolescents and young adults cared for their teeth included brushing, flossing, mouthwash, visiting a dentist, and avoiding damaging food. Interestingly, nearly all (97%) of participants reported brushing their teeth as a routine oral hygiene practice; however, only 56% reported flossing to care for their teeth, and 19% specifically reported following ADA guidelines of brushing twice a day and flossing once a day [12]. The difference between brushing and flossing could reflect a difference in perception of importance of each practice. A study examined the stability of oral health related beliefs including using dental floss at ages 15, 18, and 26 years, and found that 49% of adolescents thought flossing was not important at least once at the three ages [9]. This might explain why our study population favored brushing over flossing. A systematic review and meta-analysis has shown the benefits of flossing in addition to toothbrushing in reducing gingivitis compared to toothbrushing alone, though it lacks evidence of its effect in preventing dental caries [13]. The clinical benefits of flossing in preventing gum disease may help to formulate dental health messages to adolescents from a perspective of both cosmetic and health reasons [14].

Slightly more than half of adolescents and young adults in our study reported visiting a dentist every six months or more frequently, while 38% visited a dentist less frequently or not at all. This supports previous national statistics of dental visits by adolescents in the United States, which found that 64.9% of adults 18 and older visited a dentist in the past year [15]. The frequency of dental visits varies based on individual needs [16], though fewer dental visits could indicate limited resources or limited access to dental care. Our data showed that respondents that self-identified as non-Hispanic black or Hispanic and people who received free or reduced lunch were less likely to visit a dentist in a recommended schedule compared to their counterparts. Also, our study showed that about 10% of youth reported no dental visit at all, which could be due to barriers preventing adolescents from seeking dental care such as access to dental insurance, financial limitation, lack of adequate transportation, and social support. Infrequent or inadequate number of dental visits may contribute to a possible knowledge gap in maintaining oral health. For example, though the respondents commonly mentioned brushing to care for their teeth, it was rare for them to discuss the benefits of fluoride toothpaste, which plays an important role in preventing dental caries [17]. Also, some participants mentioned using mouthwash, whitening strips, and retainer to care for their teeth, which are more aesthetics driven than health driven. Access to dental health professionals and proper oral hygiene education may be an area for improvement in adolescents.

A total 44% of our surveyed population stated that COVID-19 had an impact on their dental health. Those who were affected by COVID-19 often reported scheduling conflicts and worsened dental hygiene. The responses to the COVID-19 impact in our adolescent and young adult samples shed a light on the psychosocial aspects of oral health in adolescence. For instance, one respondent said "We haven't gone to the dentist in a while. Also I got really depressed and it was hard to brush regularly". Another responded "Being at home all the time makes me forget to brush my teeth sometimes since I don't have as structured of a routine and don't have to for social reasons as often". Social interactions among adolescents were largely affected by COVID-19 pandemic and it may be linked with less ideal oral hygiene practice in this group, given the self-reported importance of dental aesthetics in this population.

This study has several strengths including a high response rate as well as detailed qualitative data that allowed a nuanced assessment of the rationale behind participants behaviors. The information obtained can help shape public health messaging campaigns and further inform and shape dental practitioners' approach in educating the youth about proper dental hygiene.

This study also has limitations. Participants represent a large nationwide sample of youth that opted in to a text messaging poll. However the sample is not nationally representative, which limits the generalizeability of our findings. The responses were also taken in May 2021, a very dynamic time where thoughts and opinions of COVID-19 in responses may change from the start of the pandemic to today. MyVoice is an open-ended poll designed to be only 5 questions to minimize burden on our youth participants, though this limits our ability to probe more deeply into the experiences of youth.

In summary, our study demonstrated that majority of adolescents and young adults perceived dental health to be important or somewhat important, though their oral hygiene practice may not follow ADA guidelines and the burden of self-reported dental caries is high. The COVID-19 pandemic affected adolescent's ability to schedule a dental visit and might affect their oral hygiene practice in a socially isolated environment. Gender, race, and status of receiving free or reduced lunch were associated with less frequent dental visits. Re-engagement of adolescents and young adults in dental care by dental care providers via greater access to appointments and youth-centered messaging that reinforces hygiene recommendations may help youth achieve improved dental health now and in the future.

## Author Contributions

**Conceptualization:** Long Zhang, Marika Waselewski, Margherita Fontana, Tammy Chang.

**Data curation:** Long Zhang, Marika Waselewski, Jack Nawrocki, Ian Williams.

**Formal analysis:** Long Zhang, Marika Waselewski, Jack Nawrocki, Ian Williams.

**Supervision:** Marika Waselewski, Margherita Fontana, Tammy Chang.

**Writing – original draft:** Long Zhang.

**Writing – review & editing:** Marika Waselewski, Margherita Fontana, Tammy Chang.

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
