## [Decision Letter · Decision Letter 0]

24 Oct 2022

PONE-D-22-25483Perspectives on dental health and oral hygiene practice from US adolescents and young adults during the COVID-19 pandemicPLOS ONE

Dear Dr. Chang,

Thank you for submitting your manuscript to PLOS ONE. After careful consideration, we feel that it has merit but does not fully meet PLOS ONE’s publication criteria as it currently stands. Therefore, we invite you to submit a revised version of the manuscript that addresses the points raised during the review process.

It is indeed an interesting study. However, I would suggest to respond to the reviewers comments.

We look forward to receiving your revised manuscript.

Kind regards,

Tanay Chaubal

Academic Editor

PLOS ONE

**Journal Requirements:**

"This research was funded by the Michigan Institute for Clinical & Health Research and the University of Michigan Department of Family Medicine. Funders had no part in the study design, data collection, analyses, interpretation of findings, or the decision to submit the manuscript for publication."

"TC received funding for this research from the Michigan Institute for Clinical & Health Research and the University of Michigan Department of Family Medicine. The funders had no role in study design, data collection and analysis, decision to publish, or preparation of the manuscript."

Reviewers' comments:

Reviewer's Responses to Questions

**Comments to the Author**

1. Is the manuscript technically sound, and do the data support the conclusions?

Reviewer #1: Yes

Reviewer #2: Yes

2. Has the statistical analysis been performed appropriately and rigorously? 

Reviewer #1: Yes

Reviewer #2: Yes

3. Have the authors made all data underlying the findings in their manuscript fully available?

Reviewer #1: Yes

Reviewer #2: Yes

4. Is the manuscript presented in an intelligible fashion and written in standard English?

Reviewer #1: Yes

Reviewer #2: Yes

5. Review Comments to the Author

Reviewer #1: The manuscript discusses the oral health awareness of a very important age group. The questions asked

could have included the diet of the individuals and if they had access to a dentist easily. Also, the socioeconomic status is not mentioned in the demographics. The sample size is also inadequate to make a general observation of all adolescents and young adults. The questions on COVID-19 impact lack depth and could have added information if the oral health status like bleeding gums were exacerbated at that time. Overall, not a very insightful study. Adding a larger sample size with relevant questions like the above will help us get a true picture of this population.

Reviewer #2: line 168 - in the sentence the surveyed and percieved should be seperated by a comma(,)

line 170- there is extra comma after the hygiene word, please remove.

check and go through similarly such commas are being added, please remove those also

6. PLOS authors have the option to publish the peer review history of their article (what does this mean?). If published, this will include your full peer review and any attached files.

Reviewer #1: No

Reviewer #2: No

---

## [Author Response · Author response to Decision Letter 0]

29 Nov 2022

Dear Dr. Chaubal, 

Thank you for your email dated October 24, 2022, regarding our manuscript. We greatly appreciate the suggestions and have edited our manuscript accordingly. Our point-by-point responses to the comments are detailed below. Thank you for considering our revised manuscript for PLOS ONE. 

Sincerely, 

Tammy Chang, MD, MPH, MS

Associate Professor

University of Michigan 

RESPONSE TO REVIEWERS

Reviewer #1: The manuscript discusses the oral health awareness of a very important age group. The questions asked could have included the diet of the individuals and if they had access to a dentist easily. Also, the socioeconomic status is not mentioned in the demographics. The sample size is also inadequate to make a general observation of all adolescents and young adults. The questions on COVID-19 impact lack depth and could have added information if the oral health status like bleeding gums were exacerbated at that time. Overall, not a very insightful study. Adding a larger sample size with relevant questions like the above will help us get a true picture of this population. 

RESPONSE: The following statements have been added to the limitations section to address this reviewer's concerns: 

Participants represent a large nationwide sample of youth that opted in to a text messaging poll. However, the sample is not nationally representative, which limits the generalizeability of our findings. The responses were also taken in May 2021, a very dynamic time where thoughts and opinions of COVID-19 in responses may change from the start of the pandemic to today. MyVoice is an open-ended poll designed to be only 5 questions to minimize burden on our youth participants, though this limits our ability to probe more deeply into the experiences of youth. 

Reviewer #2: line 168 - in the sentence the surveyed and perceived should be separated by a comma(,)

line 170- there is extra comma after the hygiene word, please remove.

check and go through similarly such commas are being added, please remove those also 

RESPONSE: The placement of the commas noted appears appropriate, though we will follow the guidance of the copy editor if changes are needed. 

EDITORS COMMENTS 

RESPONSE: We have added the formatting requirements that were overlooked. . 

"This research was funded by the Michigan Institute for Clinical & Health Research and the University of Michigan Department of Family Medicine. Funders had no part in the study design, data collection, analyses, interpretation of findings, or the decision to submit the manuscript for publication." 

"TC received funding for this research from the Michigan Institute for Clinical & Health Research and the University of Michigan Department of Family Medicine. The funders had no role in study design, data collection and analysis, decision to publish, or preparation of the manuscript." 

RESPONSE: The above funding statement is correct and no amendment is needed. Thank you for placing it in the Funding Statement section. 

RESPONSE: As part of our NIH Certificate of Confidentiality, even de-identified open-ended responses cannot be shared publicly without agreement to the terms of data protection for the use of such data. This is because responses may be de-identified of traditional identifiers (demographics, location, etc), however, the youth's responses themselves may include identifying information. If data is requested, the requestor can sign a data use agreement that ensures protection of our participants to receive the data.

---

## [Decision Letter · Decision Letter 1]

2 Jan 2023

Perspectives on dental health and oral hygiene practice from US adolescents and young adults during the COVID-19 pandemic

PONE-D-22-25483R1

Dear Dr. Tammy Chang,

We’re pleased to inform you that your manuscript has been judged scientifically suitable for publication and will be formally accepted for publication once it meets all outstanding technical requirements.

Kind regards,

Tanay Chaubal

Academic Editor

PLOS ONE

Additional Editor Comments (optional):

Reviewers' comments:

Reviewer's Responses to Questions

**Comments to the Author**

1. If the authors have adequately addressed your comments raised in a previous round of review and you feel that this manuscript is now acceptable for publication, you may indicate that here to bypass the “Comments to the Author” section, enter your conflict of interest statement in the “Confidential to Editor” section, and submit your "Accept" recommendation.

Reviewer #1: All comments have been addressed

Reviewer #2: All comments have been addressed

2. Is the manuscript technically sound, and do the data support the conclusions?

Reviewer #1: Yes

Reviewer #2: (No Response)

3. Has the statistical analysis been performed appropriately and rigorously? 

Reviewer #1: Yes

Reviewer #2: Yes

4. Have the authors made all data underlying the findings in their manuscript fully available?

Reviewer #1: Yes

Reviewer #2: Yes

5. Is the manuscript presented in an intelligible fashion and written in standard English?

Reviewer #1: Yes

Reviewer #2: Yes

6. Review Comments to the Author

Reviewer #1: As stated in the response to my comments, it is acceptable that the mentioned limitations such as the number of questions and the intention not to burden the youth participants of the study are valid, especially at the peak of the COVID-19 pandemic. These responses can help understand the dental/oral health perceptions of the youth better and formulate an appropriate oral health policy for them.

Reviewer #2: line 183-comma before the word and should be deleted

line 192 - same way comma after sampling method not required

line 213 and 214 if is a single sentence should be seperated by a comma and not fullstop which is placed befor the word our.

7. PLOS authors have the option to publish the peer review history of their article (what does this mean?). If published, this will include your full peer review and any attached files.

Reviewer #1: No

Reviewer #2: No

---

## [Editor Report · Acceptance letter]

6 Jan 2023

PONE-D-22-25483R1 

Perspectives on dental health and oral hygiene practice from US adolescents and young adults during the COVID-19 pandemic 

Dear Dr. Chang:

I'm pleased to inform you that your manuscript has been deemed suitable for publication in PLOS ONE. Congratulations! Your manuscript is now with our production department. 

Kind regards, 

on behalf of

Dr. Tanay Chaubal 

Academic Editor

PLOS ONE